# Real-Time Human Recognition at Night via Integrated Face and Gait Recognition Technologies

**DOI:** 10.3390/s21134323

**Published:** 2021-06-24

**Authors:** Samah A. F. Manssor, Shaoyuan Sun, Mohammed A. M. Elhassan

**Affiliations:** 1College of Information Science and Technology, Donghua University, Shanghai 201620, China; shysun@dhu.edu.cn; 2Faculty of Engineering and Technology, University of Gezira, Wad Madni 22211, Sudan; 3School of Informatics, Xiamen University, Xiamen 361005, China; mohammedac29@stu.xmu.edu.cn

**Keywords:** human recognition, integrated face–gait, few-short learning (FSL), multimodal learning, person detector, surveillance, thermal infrared (TIR) images, YOLOv3

## Abstract

Human recognition technology is a task that determines the people existing in images with the purpose of identifying them. However, automatic human recognition at night is still a challenge because of its need to align requirements with a high accuracy rate and speed. This article aims to design a novel approach that applies integrated face and gait analyses to enhance the performance of real-time human recognition in TIR images at night under various walking conditions. Therefore, a new network is proposed to improve the YOLOv3 model by fusing face and gait classifiers to identify individuals automatically. This network optimizes the TIR images, provides more accurate features (face, gait, and body segment) of the person, and possesses it through the PDM-Net to detect the person class; then, PRM-Net classifies the images for human recognition. The proposed methodology uses accurate features to form the face and gait signatures by applying the YOLO-face algorithm and YOLO algorithm. This approach was pre-trained on three night (DHU Night, FLIR, and KAIST) databases to simulate realistic conditions during the surveillance-protecting areas. The experimental results determined that the proposed method is superior to other results-related methods in the same night databases in accuracy and detection time.

## 1. Introduction

Currently, human recognition is a very active research area in the computer vision and pattern recognition community. This field has developed rapidly and is widely applied in safety applications and intelligent video surveillance solutions using technologies of facial recognition, gait recognition, tracking of human motion activity, and video human co-segmentation [1,2,3,4,5]. Human recognition is a task that identifies a person’s features in arbitrary (digital) images or videos and ignores anything else. In other words, the task of human recognition involves both the detecting of the person’s existence in the image and authenticating the identity of the human being by determining the individual (i.e., human localization) in the boundary. Identifying stationary/non-stationary objects from a video series is one of the most challenging problems in computer vision. Therefore, researchers strive to ensure that the outcome of object detection, tracking, and learning must be free from ambiguity [6]. The background information from the video should be subtracted first to detect the moving object effectively. However, modeling techniques suffer from high computing and memory costs in high-definition video, which can result in a reduction in performance measures such as accuracy and efficiency in identifying the specific object. The identification of the definite structure from a vast volume of unstructured data is a prerequisite problem to be solved. Deep learning is used to find structures in large amounts of data, which entails learning multiple layers of representation and abstraction that aid in the understanding of data such as images, sound, and text. Recently, with the development of devices, technologies, and algorithms, object detection in complex surveillance environments has become possible. In particular, the accuracy of human detection and recognition has been improved by applying convolution neural networks (CNNs) [7,8,9]. Besides, the emergence of affordable, powerful GPUs has attracted massive database designers to develop increasingly deep neural networks for all aspects of human recognition tasks—for example, person detection and feature representation classification to contribute to verification and identification solutions. However, human detection at night times remains a challenging problem. The features needed for detection are often unavailable, which makes it difficult to find human areas that are not even seen by the human eye, in addition to a large amount of visual noise. Therefore, some researchers have developed human detection systems using far-infrared (FIR) cameras [9,10,11,12,13]. They performed more processing by using modern machine learning libraries with GPU acceleration that offer optimized results at comparable frame rates [14,15].

In general, the current face recognition techniques have been shown to perform reasonably well when operating with face data obtained under controlled conditions. Besides, the experimentation with gait recognition has revealed that gait recognition systems also work well when accurate data are available for a specific set of gait variables. Therefore, when presented with face and/or gait data for an unknown person, objects will be better identified, and moving forward, will achieve a high degree of accuracy and more reliability. Several approaches have been proposed in the literature [16,17,18,19,20,21] to integrate the face (physical biometric) with gait (behavioral biometric), which verify that the combination will improve performance compared to methods that exclusively use only one of these biometrics. Although multi-biometric combination approaches have been developed, they are still considered to be in nascent stages. Nevertheless, the results reported so far are very encouraging and demonstrate the potential of these approaches for discriminating among people. However, the human detection accuracy decreases notably when dealing with objects that have large-scale changes, such as faces, changing weather, distance from the camera, and different walking conditions. Besides, it is challenging to classify a person in TIR images captured by a thermal camera, especially in images that contain more than one person, and determine people’s positions in those images. In nocturnal environments, the face classifier alone does not perform well because the high-resolution facial data is often non-available for small people, especially when in disguise. Besides, the gait classifier alone does not work well because the various walking data is often difficult to recognize in the images of missing or partially covered people, especially of an intruder. Therefore, an integration of face with gait recognition techniques was conducted in this research.

This article explores the improvement of multi-view human detection performance in real-time and solves the problems of detection and recognition at night. Thus, we propose a novel approach (named the YOLOv3-Human model) to detect varying pedestrians and track videos in a real-time webcam using their face and gait scales. The main contribution of this approach is to combine the characteristics of the face and gait into a single scale for automatically recognizing human beings effectively in TIR images under different walking conditions at night times. Therefore, a new network was proposed to improve the YOLO [22] and YOLOv3 [23] models by integrating face, gender, and gait classifiers into the single YOLOv3-Human model. This network works to obtain unique features and learn the physical (face) and behavioral (gait) biometrics for human recognition. The specific contributions are summarized in the following points: (1) the proposed methodology incorporated more detailed features from the face and gait in limitations of available thermal (TIR) imaging modalities at night by applying the FSL method to recognize the person with a limited training dataset. In contrast, the previous approaches [16,17,18,19,20] have preferred to use holistic techniques based on the silhouette in visible (RGB) imaging to identify the individual. (2) The proposed approach implements a YOLOv3 model-based method for fusing face and gait, which is better than other related methods used in the CNNs model [17,21,24,25], SVM, ANN technology [26], and YOLO model [27,28]. Besides, our approach seeks to increase recognition accuracy without combining visual (RGB) imaging features with thermal imaging features compared to related methods in the same night dataset. (3) The proposed network uses more accurate facial variables (e.g., eyes, nose, mouth, chin, etc.) to form the face signature and classify gender by using the YOLO-face algorithm [29], as well as gait variables (e.g., all spatiotemporal features, angels, speed, etc.) to make a gait signature by the YOLO real-time object detection algorithm [30]. These two algorithms are the most useful and widespread real-time detection algorithms. We also applied the face classifier in the Eigenfaces method for face and gender classification, as well as we use the person classifiers present in the modern cvlib library for gait classification. (4) This approach involves using more appropriate anchor boxes for face and gait detection, and it also obtains the function of regression loss more accurately. We evaluated our method based on the terms of accuracy rate and delay in the detection time on various night databases. Through the comparison with the other related methods on the same night datasets, our method is superior to others in accuracy rate and speed.

## 2. Theoretical Background

The ultimate goal of designing human recognition systems is to achieve the best possible classification performance for the task at hand. However, it has a critical gain when humans and machines are working together to complement each other’s strengths [5]. As the potential of Artificial Intelligence (AI) grows every day, so does corporate pressure to unleash the full power of AI by transforming entire businesses into smart systems—not just in small pockets but at scale. A comprehensive AI strategy enables one to invest in suitable systems, develop responsible practices, and design and showcase systems for a bright future. The fusion of multiple sources of evidence is likely to yield tangible benefits in terms of improving the efficiency and accuracy of the identification system [11,13]. Furthermore, fusing the data from multiple sensors makes the system more robust and reliable than single sensor-based systems. Therefore, this section briefly discusses the multimodal fusion methods as a precursor to our system framework and experimentation setup.

Machine learning has been highly successful in data-intensive applications, but it can be challenging to use when the dataset is limited [31]. Therefore, a lot of state-of-the-art methods and many smart strategies for sensor fusion using AI models have been proposed in the literature to address this problem. These methods and strategies have turned AI into action, as well providing AI with the ability to rapidly generalize to new tasks containing only a few samples with supervised information, such as few-short learning (FSL) methods [32] and multimodal machine learning technology [33]. FSL is a technology also referred to as low-short learning (LSL) in few sources. This technology is a type of machine learning problem where the training dataset contains limited information [34]. When FSL is applied to deep learning approaches, it can be used to train models on the only source of information: images, video, text, and audio. [35]. FSL methods have been categorized from three perspectives [31]: (i) data-based approaches, which uses prior knowledge to augment the supervised experience; (ii) model-based approaches, which uses prior knowledge to reduce the size of the hypothesis space; and (iii) algorithm-based approaches, which adjusts the search for the best hypothesis in the given hypothesis space based on prior knowledge. The common practice for machine learning applications is to feed as much data as the model can take. As it is in the most network (e.g., Siamese, Triplet, and Matching Networks [36]), algorithms (e.g., MAML [37], FOMAM, Reptile, and LSTMs [38]) and computer vision applications (e.g., Facebook AI Research [39]), feeding more data enables the model to predict better. However, FSL aims to build accurate machine learning models to predict the correct class of instances when a small number of samples are available with a smaller training dataset. Moreover, the experience of multimodal deep learning technology [40] refers to the involvement of several human senses—visual, auditory, reading and writing, and kinesthetic (VARK)—in processing information during learning to understand patterns and remember more. By combining these patterns, experience learners can combine information from different sources to create a diverse learning style [41], as shown in Figure 1.

However, VARK is still a valuable model that should remain at the top of attention when creating diverse learning content to engage the learner. Research has shown that learning in multiple modalities enhances the understanding of knowledge, which confirms the need for a multimodal fusion learning strategy in machine learning and deep learning systems. From a quality point of view, multimodal learning creates a more exciting and inclusive learning environment for programmers. They are not obliged to train the models in a conventional manner, which increases the accuracy of the recognition in the models of learning and knowledge. Besides that, the multimedia fusion learning technology contributes to differently integrated types of fun and exciting media in different fields to improve multimedia learning systems [42,43,44]. From videos to interactive infographics, multimodal learning strategies can take advantage of technology and digital learning tools that users love to use in the future. Surpassing human-level performance propelled the research in applications where different modalities amongst language, vision, sensory, text play an essential role inaccurate predictions and identification [45]. Several state-of-the-art approaches in multimodal fusion employing deep learning models are proposed in the literature, such as approaches presented by F. Ramzan et al. [24], A. Zlatintsi et al. [25], M. Dhouib and S. Masmoudi [26], Y. D. Zhang et al. [27], C. Devaguptapu et al. [28], and P. Narkhede et al. [46]. The purpose of these approaches is to enhance the multimodal fusion method with which the objects can be efficiently detected from static images or given video sequences with the preferable use of the deep learning library. Despite their outstanding performance, the opaque, complex, and black-box nature of the deep neural nets limits their social acceptance and usability. This has sparked a search for model interpretability and explainability, which is especially important in complex tasks involving multimodal AI methods.

To improve the efficiency of the multimodal systems [19,47,48], first, a multistage combination method is adopted, whereby different subjects can be classified by a less precise classifier, which passes a smaller set of potential candidates to a more accurate classifier. In addition, the decisions of different classifiers are combined directly via SUM and PRODUCT rules [49,50]. For example, in the fusion face and gait recognition system, a decision-level fusion is a special case of data fusion that combines the results of the face and gait recognition algorithms. Thus, two multimodal systems (i.e., based on source face + gait) are fused. Specifically, for the first subspace (l), given the normalized combine gait scores (*S^l^ gait*) and the normalized combine face scores (*S^l^ face*), the updated scores of the fusing face and gait (*S^l^ face* + gait) corresponding to the decision fusion method [17,18,19,20] are defined in Equation (1):(1)$$S^{l}face+gait=\omega \;S^{l}face+S^{l}gait$$

## 3. Proposed Approach

This article aims to present a novel approach called the YOLOv3-Human model, which enables us to recognize a human in different situations in real-time night environments. To achieve this goal, we developed a multi-view normalization approach by integrating face–gait recognition techniques into a single approach, thus improving human recognition in TIR images or night video sequences (Section 3.2). First, the features of several variants of a face sample and features of several walking conditions of the gait sample are fused individually using the averaging function. Then, the individual face and gait features are combined to form a single-feature vector sequentially, thus increasing the information fusion from different sources as well as contributing to decision-making. In our case, we reduced the size of the hypothesis (H) space using prior knowledge, and then we chose a suitable algorithm (YOLOv3) that alters the search for the best hypothesis in the given hypothesis space using prior knowledge. Our methodology uses prior knowledge in the FSL method to constrain the complexity of H, which results in much smaller hypothesis spaces. Furthermore, this approach learns to solve multiple related tasks jointly; hence, the task parameters are constrained by the other tasks.

### 3.1. The Overall Procedure of the Proposed Methodology

Figure 2 illustrates the overall procedure flowchart of this research, as shown in following below:Input: A TIR image or video (human or/and background) of the pedestrian is captured by the thermal camera with different views (height × width pixels) to render virtual views of detection, tracking, and recognition.Normalization: The variable size (1920 × 1080 pixels, normalized human or background) of the input image is much greater than the size (416 × 416 pixels) of the input image for the proposed YOLOv3 network that uses a square size (N height × N width pixels), as in other CNN methods [10,51,52]. Moreover, the details are blurred, the tone is dark, the noise pollution is large, and it is difficult to extract the contour. Therefore, we need to resize the input image and normalization process. For a normal human area in TIR images, the body’s height is far greater than its width within the specified database. Hence, when performing square-shaped size normalization, the TIR image is stretched in height and width, which distorts the human area and makes it challenging to detect features accurately. Therefore, the size normalization is performed by bilinear interpolation in order to obtain a fixed size (416 × 416 pixels) of the TIR image. The person’s size changes are compensated through the normalization size when people are close to or further from the camera by applying the zero-center method [10] to perform brightness normalization on that image.Optimization: After normalization, the TIR image is optimized by passing through into OTI-Net, which uses the histogram equalization method and dynamic threshold difference method [53] to adjust the brightness. If the optimized TIR image is square, then it will be used as the input for person detector manner (PDM-Net) to verify the human detection process.Detection: In the PDM-Net, the image is passed via a single CNN of the trained new proposed YOLOv3 network without horizontal stretching, and will include a small background space around the human area. The canonical viewpoints were determined by examining the 3-D structure of the face, gait, appearance (texture), and motion of a moving person; hence, the detection of the face and gait features is more accurate.Classification: The proposed network classifies the image into only two classes, as either a human (1: person) or background area (0: nonperson) by using integrated face–gait classifiers according to tasks of the YOLOv3-Human network. In contrast, the network classifies a person who requires a more complex structure.Output: the method combines the human image and background as an original input in order to obtain results for real-time human recognition and identification, which significantly reduces the training and detection time.

### 3.2. Integration of Face and Gait Recognition Technologies

This subsection presents the design of a novel approach that integrates the face and gait recognition technologies into a single model in different night situations. Therefore, we applied two complementary solutions (multimodal learning) for human recognition: face classifiers and gait classifiers. The overall multimodal learning task is divided into three steps [28]:(1) learning the individual features, which contain at least two information (e.g., face and gait) variables; (2) processing the information fusion; and (3) testing the model trained on the combined information. Looking at the steps involved in multimodal learning, we carried out in more detail:Representation of modalities: The essential first step is learning how to represent the inputs and summarize the data to communicate the different modalities. The heterogeneity of multimodal data makes some challenges to create such representations.Processing or Translation: A second step is to process and convert data from one modality to another. Although the data is heterogeneous, the relationship between the two modalities is often ambiguous or open-ended. Therefore, there has to be a direct relation between (sub) elements from two or more different modalities. For example, in the TIR image translation, the features extracted from gait images are in the form of finer details, such as environmental surroundings, edges, shape, angels, and steps, while corresponding features extracted from the face images are in the form of distinct points.Fusion and Co-learning: A third step is to combine information from two or more modalities to improve our network’s predictive ability for human recognition. For example, in order to recognize a person, a visual description of body movement is integrated with facial features to predict the person’s class under different walking conditions. The information coming from different modalities has varying predictive power and noise topology, with potential data loss or missing data in at least one of the modalities.

For fusing the face and gait classifiers, a decision-level fusion strategy (Equation (1)) was followed where the best matches for the face classifiers are passed on to the gait classifiers, which determines the unknown person, as shown in Figure 3.

First of all, TIR images or video frames were clipped depending on the human body’s segmentation. The AI divided these clips into face image sequences and gait frequency sequences. Next, an enhanced face image (EFI) was generated from the images as face templates, as well as construction of the enhanced gait image (EGI) from the video frames as gait templates. These templates are tested by a separate component with feature analysis based on the face and gait templates obtained from all training videos during the training procedure. As a result, the more complex features are obtained during the training process, as well as the transformation matrices that later make up the face and gait variables in the probe and gallery sets. Each test video is processed during the recognition process to create both face and gait templates and then be transformed by the previous transformation matrices to extract facial and gait variables, respectively. The target video’s variables (or features) are compared with the gallery images’ variables presenting in the dataset during the testing process. Then, the fusion technique is used to integrate face, gender, and gait classifiers to enhance recognition efficiency. The reasons for introduced gender recognition in addition to face recognition is because gender recognition [54] is an essential task for many applications that need personal, reliable, and ethical systems, as in, for example, applications of human–computer interaction and computer-aided physiological or psychological analysis. In addition, the AI contains a wide range of information regarding the distinctive difference between males and females. Thus, this approach can be contributed to detecting the perpetrators of security incidents in human identity (HID), safety, and surveillance systems [43] in order to disclose gender-based violence (GBV) [55].

To fuse the face classifiers, the YOLO-face algorithm [29] with the Eigenfaces method [56] is used to address the detection problem of various facial scales, as well as a Bayesian-inference-based classifier [57] is employed for face recognition and gender classification. The YOLO-face algorithm aims to obtain the features, i.e., facial variables (e.g., eyes, nose, mouth, chin, etc.) from the face images. The facial features are extracted by the proposed network (new tiny-yolov3) by combining multiple tasks during the training process. Multi-task learning is a framework for training in a typical architecture of various tasks. Layers will learn a joint generalized representation at the beginning of the proposed network and prevent over-fitting to a particular task that may involve noise.

To fuse the gait classifiers, the YOLO real-time object detection algorithm and the Hidden Markov Models (HMMs) method [58] are used to address the detection problem of various gait scales (e.g., some spatiotemporal variables with angle trajectories). These variables form the gait signature, such as the following: (1) anthropometric signs, the lengths of the skeleton segments, and human height; (2) signs of relative distances obtained from the difference of coordinates of various points of the skeleton (joint angles); as well as (3) signs of movement based on the displacement of the skeleton node between two adjacent frames (walking speed). The model applies the YOLO real-time object detection algorithm to form the person’s binary signature in the TIR image, which uses various gait (or walking) features as the most important datasets for human recognition.

These two algorithms (YOLO and YOLO-face) and methods (Eigenfaces, Bayesian, TMMs, and HMMs) are tested on different night databases. In addition to implementing two types of human recognition strategies using depth data in TIR images and working only with the night video sequence. The Sum-Square Error loss function is used, leading to regression loss to make the detection algorithms more systematic. Finally, the similarity between the sensor and display sequences is measured by computing the first keyframes extracted from the corresponding sequence pairs and normal-sized frames for human identification.

### 3.3. Feature Extraction and Recognition

In this work, we applied unsupervised feature learning for individual modalities—e.g., face, gait, or tracking walking in real-time. Therefore, the weighted combination of the subnetworks (e.g., YOLO-face and YOLOv3) is taken so that each input modality has an acquired contribution towards an expected output. As a result, the proposed model (fusion of face and gait classifiers) enables a more useful inclusion of facial and gait features from the same TIR image source than other related methods. Moreover, the model architecture can be chosen for different modalities according to need—e.g., a YOLO-face algorithm for face data or a YOLOv3 algorithm for gait data. Besides that, we then combine the features and pass the product to the final fusion face–gait classifier by aggregating the modalities (Figure 3).

In face recognition and gender identification, they are operating in both or either situations [59]: one-to-one (1:1) match and one-to-many (1: M) match. On one side, the verification (gender identification: male or female) can be defined as the 1:1 match that compares the features of the query face against the features of the target gender whose personality is being claimed and decided (yes: male or no: female). On the other side, face recognition is a 1: N match that matches a query face against the faces available in the database and offers a ranked list of matches. Figure 4 shows how to identify and verify the face in the face recognition module [60]. In this figure, a TIR image (physical or behavioral) and the pedestrian’s video frames captured by the thermal camera during night conditions are inserted into the input model. In GPU computing, the detection algorithm determines whether the face is presented in the image source or not. If it happens, each face’s position and location in the image is determined and it directly captures information about the shapes of faces and describes the facial variables of surfaces, such as the eye hole, nose, and chin curves of each individual. The Eigenfaces algorithm is used to pick out specific distinctive details (or information) about a person’s face. Besides that, the feature-based YOLO-face method is applied to extract (and measure) distinctive facial features as well as other fiducial marks and then compute the geometric relationships among those facial points. Thus, it reduces the input facial image to a vector of geometric features. These details are converted into a mathematical representation compared with the other details of faces collected in the night database. The particular face data is called a face template (or sample) and is distinguished from a photograph. It is designed to use only some details to distinguish one face from the other (1: M match method). Finally, statistical pattern recognition techniques are applied to match probe faces with templates stored in the database.

In gait recognition and body measurement, we assume the human body is in the standing, walking, running, or driving position, and the human body comprises five segments (e.g., head, torso, two hands, and two legs). Therefore, the walking features were extracted from the body image obtained from a thermal infrared camera according to those segments to find individuals. Therefore, the distinction of walking direction (left, right, in front of, or behind the camera) is one of the main features of walking [61]. First, if the head is detected, the change detection rate between the images becomes large and the trend is taken towards a larger one. In contrast, when the header is not detected, the image is determined by changing the body motion in the extraction region [42]. Therefore, this method is focused on the torso [62] in particular because it has a central role in human recognition. Figure 5 shows the sequential stages of extracting the human body and gait features for recognition. First, the body of a pedestrian is divided into several edge points (or segments) representing the parts of the body. Second, the four parts (two edges of the hands and the legs) are symmetrically connected with the torso section (called the vertical axis) that passes from the middle of the body in a particular position. Third, the width of hands and the length of legs is converted from edges to lines and assumed to be equal (2:2). Fourth, the width and length of body segments (hands and legs) are used together with the position area to classify lines based on slope. Fifth, the model searches the segments of the body in the human image to obtain more complex features (or variables). Sixth, these characteristics (position, length, width, and area of the clip (or segment) helps the YOLO and the HMMs algorithms to detect a human body’s existence in the images. Seventh, the characteristics of each part of the body are extracted from the pedestrian gait of the test image based on the previous stages, and then transform it to acquire the amount of characteristic walking to compare it (or decision process) to previous features stored in the database. Finally, the results of different parts are combined into one score as a final result: human recognition.

In our case, the gait recognition algorithm yields the results from a distance measurement, while the face recognition algorithm yields the results from a match score or probability. Therefore, the model describes the face and gait with different variables and recognizes individuals through the integration of vectors estimated from the sequence of variables. After the updated scores, a majority vote is introduced for the final classification decision. Besides that, the Template Matching Methods (TMMs) [63] was used to combine the fundamental analysis-based transformation with the Eigenspace transformation to select the person class for recognition.

Finally, when the less reliable soft biometric traits (face, gender) with low weights were fused, a significant performance gain was achieved. Although the face is less reliable in the real-time human recognition system due to low resolution or the presence of other covariates, it may provide some complementary information for gait when using it as the ancillary information with the low weight assigned. Therefore, we use the face as a soft biometric trait with its low score weight assigned to the corresponding evaluations.

### 3.4. Proposed Network Architecture

The proposed network architecture contains the optimized thermal image network (OTI-Net), person detector manner (PDM-Net), and person recognition manner (PRM-Net), as shown in Figure 6. This network optimizes the TIR image through the pass-over OTI-Net. Next, the task of PDM-Net provides more accurate features (face, gait, and body part) of the person from the enhanced images to detect a person class. Besides, the person’s image is classified and then recognized as a human according to his/her face or/and gait using the PRM-Net.

The OTI-Net architecture is designed based on the CNNs, which have eight-layer convolution (Conv) and eight-layer deconvolution (DeConv) with the down-sampling layer, as shown at the top of Figure 6. In the OTI-Net, we chose 416 × 416 × 3 as the CNNs input. The asymmetrical structure occurs with serial communication during Conv and Deconv layers. When an image passes through the Conv layers, the size of the image is reduced two times. However, the size doubles each time the image passes through the DeConv layers. Therefore, after Conv and Deconv, the input size is still 416 × 416 × 3. Besides that, the down-sampling layer that interpolates the OTI-Net is connected with the PDM-Net input.

The PDM-Net architecture is designed by updating the original convolution layers of the YOLOv3 backbone [64], which is mainly composed of the feature extraction layer (Darknet53), and detection layer (face and gait detectors). This architecture comprises the convolution layer of the Darknet53 [64,65], Batch Normalization (BN) Layer, concatenate layer, and Relu activation function, as shown in Figure 7.

First and foremost, the enhanced image (EFI or EGI) is passed through Darknet53, where the maximum stride of the network is 32, causing the input image to be a specific size (416 × 416 × 32). Next, this image is optimized by the detection layer so that the PDM-Net can identify individuals either by recognizing a face or gait at night. Therefore, a new alternative version of the tiny-yolov3 network has been proposed, which contains seven convolutional (Conv) layers and six pooling (Pool) layers, with a difference in its modular structure, as shown in Figure 8.

The proposed network uses a single CNN for processing the TIR image and divides this image into regions, and then predicts the bounding boxes to obtain each region’s probabilities. This network applies many layers (residual block) with a slightly more complex shape, but it is still just a regular convention. Besides, we added a new (Conv block) layer to the start of the standard YOLOv3 network, so it preempts the standard network layers. Therefore, the overall layers in the proposed network contain the input layer, deep convolution (new tiny-yolov3) layers, two up-sampling layers, two concatenate layers with Conv block, and three classification layers.

In the network, the first six deep convolution layers (Conv1 to Conv6) have a 3 × 3 kernel size, followed by the max-pooling with a 2 × 2 kernel, and no fully connected layer. The very last convolutional layer (Conv7) has a 1 × 1 kernel size, and it reduces the size of a 13 × 13 × 1024 shape to the desired outputs. Therefore, the network ends with three (124, 256, 256) channels for three grid cells. These channels have to contain the data for the bounding boxes and the class predictions. Each grid cell predicts data elements described by five bounding boxes and confidence scores, where (bx, by, width, height) defines the parameters of the bounding box’s rectangle. The (bx, by) represents the center positions of the object label, and (width, height) represents the width and height of the object label. The confidence score refers to the probability distribution over only one object (person) class. We need to compute the final scores for the bounding boxes and throw away the ones scoring lower than 30%. The input image’s size is resized to 416 × 416 pixels, which goes through the convolution network in different three passes and comes out the other end as (A: 13 × 13 × 124), (B: 26 × 26 × 256) and (C: 52 × 52 × 256) tensors describing the bounding boxes for the grid cells. Despite the structural similarities of input and output, the three scales (A, B, and C) are used for classification in the PRM-Net, where “A” is to classify a small person, “B” is to classify a large person, and “C” is to classify a face with gender. All boxes are divided into three anchor boxes, and one of them is taken as the final detection result. Finally, the testing protocol is applied from a custom dataset to another reference dataset to verify samples’ portability trained on the model. Therefore, we will see a recent update to the YOLOv3 method using original researchers to recognize pedestrians at night.

## 4. Experiments

This section presented the experiment of the implementation. We have prepared data to train a custom face–gait detector manner. To teach a deep model, it is necessary to set up an experimental environment for the learning model and choose an appropriate nighttime human database with annotation. There are two aspects to improve the detection accuracy rate and increase the delay in the detection time: improving the image quality and selecting a detection algorithm. The TIR images in the night dataset were optimized and then applied the YOLO algorithm to train the sampling set. Besides that, the data set has two aims: one is to train the YOLOv3 model; the other is the component of the YOLOv3-Human model. The pre-trained model will be run by the YOLOv3-Human inference and tested on the test set.

### 4.1. Experimental Environment Set Up

To set up an experimental platform environment, we need to install the Python program on GPU computing. The GPU computing runs under Windows 10 Professional Operating System with an Intel^®^ Core (TM) I5-6600 CPU @ 3.30 GHz, 3.30 GHz, and 16.00 GB RAM. We recommend using the Nvidia GPU Acceleration Library (NVIDIA GeForce GTX 1070, 1, CUDA 10, CUDNN7.4). It requires installing available libraries such as cvlib 0.2.2, Open CV 4.2.0.34 (or above), TensorFlow GPU version 1.13.1 (or above), NumPy, Progressbar, Requests, Pillow, Keras, imageio, and utils.

### 4.2. Dataset

In our experimental studies, we used the recently released custom DHU Night dataset [66] for face and gait recognition of large persons, and also used two reference night databases, e.g., the FLIR ADAS dataset [28] and KAIST dataset [67] to recognize a small person and multiple people. The DHU Night dataset [66] is a pedestrian dataset captured at Donghua University (DHU) during nighttime. This dataset collects the thermal images of up to 10 persons and their coordinates in the image. All videos in the dataset are recorded in different conditions at night—regular (with glasses, cap, and short or long hair) and running in different positions and ranges (angles) from the scenes’ thermal camera. The dataset provides 3.5 k images processed into a usable dataset. We split this dataset into 3 k for training and 0.5 k for testing. We have used the LabelImg Open-Source software to annotate video sequences in YOLO and PASCAL VOC formats. They show some example images from the dataset in Figure 9.

The FLIR ADAS dataset [28] consists of 9214 images with bounding box annotations. Each image in the dataset has a 640 × 512 resolution, captured by the thermal infrared (FLIR Tau2) camera. A number of 60% of these images were collected during the day (RGB images), and the remaining 40% were collected during the night (TIR images). In our methodology, we used only the TIR images from the dataset in all the experiments. In addition to using training and test splits provided in the dataset contains only one (Person) class. Figure 9 shows the examples of thermal images from the FLIR ADAS dataset.

The KAIST Multispectral pedestrian benchmark dataset [67] contains around 95 k images with annotations comprising only the Person class. The data is captured in Korea during the day (RGB images) and night (TIR images) using a FLIR A35 micro-bolometer LWIR camera with a resolution of 320 × 256 pixels at 8-Bit. These images are then up-sampled to 640 × 512 in the dataset. Finally, the images are split into 50 k images for training and 45 k images for testing. We choose the TIR images for all experiments to provide the training and testing splits containing only the Person class, thus improving the detection in the absence of paired training data. We show sample images from the KAIST dataset in Figure 9.

### 4.3. Training

This stage aims to learn the proposed YOLOv3-Human model for person identification by applying the face–gait detector. First of all, the training configuration file is set up as follows: (1) both the first and second stages were set to 10 epochs and warmup was set to 2 epochs; thus, 20 × 30 is the total number of epochs; (2) the train input size was set to (320, 352, 384, 416, 448, 480, 512, 544, 576, 608), with a batch size of six; (3) the strides were set to (8, 16, 32), where anchor per scale was 3 and IOU loss thresh was 0.5; finally, the learning rate initialization was set to $1\times 10^{-4}$- and the learning rate end was $1\times 10^{-6}$.

After setting up the configuration (conf) file, the model is directly pre-trained with the RGB COCO and Pascal VOC databases to make initially learned weights. We have used the RGB pre-trained YOLOv3-Human model for validation. Besides that, we trained the model on thermal (or TIR) images according to the proposed network architecture to obtain the best weight in the current configuration, as shown in the following steps:Imbalanced dataset: Only one object class (Person) can be observable enough with annotations by the Up-sampling layer.Custom Anchors: The anchors for a custom dataset were recalculating based on the dataset’s size distribution, thus helping our network to achieve faster convergence and better AP and mAP scores.Pre-trained Convolution Weights on Imagenet: The network takes less time to converge and obtains a better mAP score in the first calculation.Channel Number: We pre-trained the network on three-channel gray-scale images.Optimizer: The Darknet53 framework [68] was used for the feature extraction process, which is momentum. Darknet53 is an accumulation of movement because it can decay, weaken, and degrade the weights in order to obtain typical features.

Besides that, necessary files were created related to the data provided by the Darknet53 framework to carry out the training process, such as:classes.txt file—The file was used earlier with LabelImg and contained the object categories.train.txt file—Darknet expects a text file that lists all the images used for training and validation. 60–70% of the total dataset has been allocated to training, and we kept the rest for testing.test.txt file—the file contains all images which will be used for testing.traffic_test.txt file—the file contains all images used for the traffic test.

Furthermore, these files (train.txt, test.txt, and traffic.txt) were created and saved in the project directory in a convenient location. To form data, it keeps the YOLOv3-Human model and creates a text file containing the 3500 images labeled as a total. In face recognition, we used 1000 images for training, 800 images for validation, and 200 images for testing. In gait recognition, 3000 images were used for training and validation, and 500 images for testing.

With training the YOLOv3-Human model in advance, the trained weights for thermal imaging were used to improve the results according to adjustments in the threshold parameters. Thus, the model can learn features (face and gait) from the custom sample set to detect only the Person class in TIR images, video, and real-time webcam; we used the same procedure for teaching the model to detect small and multiple people from TIR images in the other night databases. Hence, the detected person’s information can be helpful for human recognition in safety and night observation fields.

## 5. Results and Analysis

This section has tested the proposed approach (named YOLOv3-Human model) on different night databases to detect and recognize individuals according to the integrated face–gait method, in addition to explaining the results obtained by using the YOLOv3-Human model with some test cases. The test evaluation indicators primarily use the accuracy rates of the Precision/Recall (PR) curve (e.g., AP scores of Mean Average Precision (mAP)) and speed or delay in detection time (ms). We explained the results of every test case using the screenshots of the output provided by our approach and compared it with other related methods. For integrated face and gait classifier tests, we first specify the image/video path in the test file and then run the python code through GPU computing to show results in the window. Next, the network looks at the image just once and then splits it into a 13 × 13 cell grid. Each cell predicts five boundary boxes to classify a person according to facial and gait features. The bounding box refers to the rectangle surrounding the person’s class. If this rectangle contains facial features, the model classifies the face and gender; otherwise, it is classified as gait based on general body segments’ features. It also generates a confidence score that tells us how likely a person is to be included in the expected bounding box. Finally, the model can identify one or more people in the image, and whether the people in an image appeared in small or large-size subjects.

### 5.1. Test with a Custom DHU Night Dataset

This test evaluated the YOLOv3-Human model on the custom DHU Night dataset for human recognition. The experimental result shows the type of object (Person), sex (Gender: male or female), and confidence score (in %) with a bounding box, only if the box shape is good. The boundary boxes are expected to appear around the person when the confidence score is higher than 30%; then, the fatter box will be drawn. Next, the bounding boxes’ confidence and the class predictions are combined into a final score to inform us of the probability of the bounding box containing a particular object type.

#### 5.1.1. Face Classifier Tests

We constructed two integrated face classifiers for face recognition by combining the Eigenfaces-based face classifier and the Bayesian-inference-based gender classifier (described in section II.B). In our case, we tested the two integrated classifiers using the gallery and probe sets containing all subjects. The face recognition results for both classifiers are shown in Figure 10. After the trainer face classifier on the DHU Night dataset, the inference was completed on each image, and then the accuracy rate (99%) at 20*30 epochs was presented on the console, as shown in Figure 10a. The total number of recognized, unrecognized, and aligned images are shown in Figure 10b. Since both integrated face classifiers use the same facial features; the difference between them directly follows the difference in their face classifiers’ performances; for example, the Eigenfaces classifies the type of object’ class, as well as the Bayesian classifies the gender.

To test the face recognition with the gender classifier, we choose the TIR images or real-time video frames from the test set as the input to create new sample images (or temples) for facial recognition. For each video frame, only one person is seen. Initially, the frame is empty. When the person enters the frame, the software correctly detects the existing person’s face and draws a rectangle around the face. After detection, the software correctly recognizes people and continues to track their faces and correctly label them as male or female. The screenshots of the model’s output for six videos are shown in Figure 11. The model is trained to recognize him and label him as a male, as shown in Figure 11a–c, respectively. Similar to videos in Figure 11d–f, they were trained based on the model to recognize her and label her as female.

As a result, gender recognition, in addition to facial recognition, explores the possibility of enhancing the results obtained from thermal imaging readings recorded from the different subjects that target the gender prediction task (male and female). This paper demonstrates that facial and gender-based differences in multimodal deception detection can be measured when the people are wearing hats, glasses, and scarves, in addition to describing and showing how the two genders achieve different detection rates for different individual and combined feature sets with an accuracy rate of 99% overall. Our results allow us to make interesting observations concerning the differences in the multimodal detection of deception in males and females.

#### 5.1.2. Gait Classifier Tests

The gait classifier was tested using the gallery and probe sets to recognize the pedestrians in TIR images and real-time videos, and the accuracy rate (Note: mPA = AP score, because the model detects only the Person class) for the gait classifier trained on the DHU Night dataset is shown in Figure 12. This classifier takes 7 ms as a delay in detecting a person with a large size in the TIR image.

Since data for specific gait parameters (or variables) will be available in real-world application scenarios, two different gait classifiers were constructed to test the proposed approach using the following two sets of variables:Tests with spatiotemporal variables

Spatiotemporal variables are individual or combined when recognized by the gait classifiers. All combination variables were then taken with the highest recognition rates. The results of the better-performing variables are depicted in Table 1. These results show that the recognition rates for three variables (i.e., step time, cadence, and stance phase time) are better than those based on individual variables but worse than those based on combining all the variables with a smaller gallery size with five subjects. All spatiotemporal variables’ recognition rates are consistent with the recognition rates obtained for individual variables with an extensive gallery of 10 subjects. The overall recognition rates for the gallery size with 10 subjects are lower than those for the gallery size with five subjects for testing. However, it should be noted that this increase in performance corresponds to an increase in exhibition size observed in tests with five subjects or fewer individuals. The above table shows that the low-performing variables contribute to the discriminative information contributed by the high-performing variables.

Tests with angle trajectories

The angle trajectory variables are recognized by applying the gait classifiers with these variables individually using lower- and upper-body angle trajectories, as depicted in Table 2 and Table 3, respectively. The right side (R) of the hip angle trajectory is the best performing variable overall, yielding a recognition rate of 100% with five subjects and 89% with five subjects using lower-body angle trajectories (Table 2). On the other hand, the left side (L) of the shoulder angle trajectory is the best-performing variable overall, yielding a recognition rate of 83% with five subjects and 41% with 10 subjects using upper body angle trajectories (Table 3).

Nevertheless, the recognition accuracy rates were obtained using two combinations of lower- and upper-body variables compared with the accuracy rates of all angel variables, as shown in Table 4. Thus, the results were presented in real-world conditions, where twice the weight was given to some low-performing variables, such as lower- and upper-body angle trajectories. However, the recognition rate based on all the parameters’ combinations is worse than the rates based on some individual parameters. This is not surprising since a combination of many parameters is being taken, many of which are not yielding good results individually, and the discriminatory potential of better-performing variables is nullified by several low-performing ones.

The following observations made based on the above results:When only the lower-body angles are combined, the recognition rate is worse than that obtained for combining all parameters in a gallery size with five subjects (95.83% vs. 96.50%) and with 10 subjects (87.82% vs. 89.40%).When only the upper-body angles are combined, the recognition rate is worse than that obtained for combining all parameters in a gallery size with five subjects (91.67% vs. 96.50%) and with 10 subjects (82.69% vs. 89.40%).The recognition rate for the combination of all upper-body angles is also worse than that for all lower-body angles in a gallery size with five subjects (91.67 vs. 95.83) and with 10 subjects (82.69% vs. 87.82%). These results show that the lower-body angles are more useful for recognition than the upper-body angles.When only the lower-body angles or only the upper-body angles are combined, the result for the recognition rate is worse (instead of better) than that obtained for the combination of all parameters.

Figure 13 shows the gait recognition results trained on the DHU Night dataset to classify only one object (person) class from TIR images or real-time night video frames using the DarkNet53 framework. We chose three different appropriate clips for testing, such as regular walking, backside walking, and the person appears in incomplete conditions. The detection algorithm looks at the TIR image only once; if it detects someone in the image, the trained model predicts the person’s boundary boxes and then observes the screen’s output. We show the detected person on the green rectangle’s bounding box with a confidence score of detection accuracy to obtain the human recognition reliability.

### 5.2. Test with Standard Datasets

We tested and calculated the accuracy rate in the proposed model’s experimental test on the FLIR ADAS and KAIST validation sets. The recognition accuracy rates (AP score) are shown in the PR curve results, Figure 14. This model has a delay in detecting a person in the image (8~10 ms).

Figure 15 shows the proposed model results on the FLIR ADAS and KAIST datasets under different walking conditions with various distances from a camera. The model has the true positive (TP) detection of a person in the different (in front of, right-side, left-side) walking conditions on TIR images captured by the thermal camera from the outdoor environment. The results are shown in Figure 15a–c,e–g, respectively. The person is also correctly detected in the model while in the backside-walking conditions; look at the right side of Figure 15 and left side from Figure 15b. In contrast, the model had TP detection in both bike- and bicycle-driving conditions, as shown in the left-hand images of Figure 15a,g, and the right-hand side of Figure 15b,d. Interestingly, in the same distance, the model failed to detect a person when the person was driving the car, Figure 15d. The model also failed to detect a tiny person who was far away from the camera, as shown in Figure 15h. Finally, we note that changes in the people’s behavior (the person in front of, right-side, left-side, backside from the camera) and activity (driving, hiding, and running) under the changed distances (somewhat) did not significantly affect the TP detection results for small-size of persons in our model.

### 5.3. Comparison with Other-Related Methods

There are three kinds of purposes for the YOLO-Human model: face recognition, gait recognition, and person detection with small-size and multi-views. In contrast, the main purpose among other related models is either to detect or recognize people in thermal infrared imaging depending on the individual method to obtain one of the biometric features (e.g., face, gait, or body, but not both). Therefore, the proposed model was compared to other related methods to measure the performance of human detection and recognition at night depending on the fusion of the face and gait method, as shown in Table 5. The results of the YOLOv3-Human model based on the fusion of face and gait biometrics were outperformed by other related results obtained by the individual biometrics recognition models, e.g., TIRFaceNet model [66], YOLO model [69], and MMTOD model [28] on the same night databases. The YOLOv3-Human model achieved an accuracy rate of 99% for face and gender recognition, which exceeds the accuracy (89.7%) achieved by the TIRFaceNet model for individual facial recognition evaluated on the same DHU Night dataset, in addition to achieving an accuracy of 90% for gait recognition with large-size subjects. Moreover, the model has a delay in a detection time of 7 ms to detect a person in thermal imaging.

Besides that, the proposed model has a true positive (TP) for detecting small and multiple subjects on both FLIR and KAIST datasets. This TP means that the model is able to detect all people in the images with their different poses (Figure 15) compared to the other individual (YOLO and MMTOD) models on the same datasets. On the FLIR dataset, the AP score (67.54%) of our model exceeded the AP scores achieved by the YOLO model (29.36%) and the MMTOD model (54.69%). The model also outperformed the accuracy of the MMTOD model (49.36%) on the KAIST dataset, which has an accuracy rate of 65.01% to detect very small (or tiny) people. Finally, with the above results, we show that the proposed methodology achieves the highest accuracy rates for identifying a person in the different night databases, especially containing large-sized subjects, and the model has a high score of TP with small and multiple subjects.

## 6. Conclusions

This paper has successfully integrated face and gait technologies to automatically recognizing human beings in TIR images and real-time video frames at night. Individual face and gait classifiers were composed and then combined to complement face and gait in different scenarios at night. Two different classifiers of face were operated on low-resolution facial images, while different gait classifiers used the dynamic gait features extracted from motion-capture data. In all cases, face and gait classifiers worked in the same model to identify individuals. We used the YOLO-face algorithm with the Eigenfaces method and the YOLO algorithm with the HMMs for face, gender recognition, and gait recognition, respectively. To combine the results of the face and gait recognition algorithms, the decision-level fusion method is used. In all experimentation tests, the YOLOv3-Human model was trained on the DHU Night, FLIR, and KAIST databases. As a result, the use of the fusion of the face and gait method demonstrated that the results were more efficient in multimodal biological systems. We also demonstrated that the combination of scores obtained by individual algorithms improved the recognition rates. Moreover, the results of the integrated face and gait method in human recognition at night outperformed the other results-related methods, which are only based on face and gait recognition individually. Besides that, the proposed approach has increased the rate of recognition accuracy while maintaining a fast detection speed. Finally, we recommend using this approach in surveillance systems to detect intruders with high efficiency at night.

## Figures and Tables

**Figure 1 sensors-21-04323-f001:**
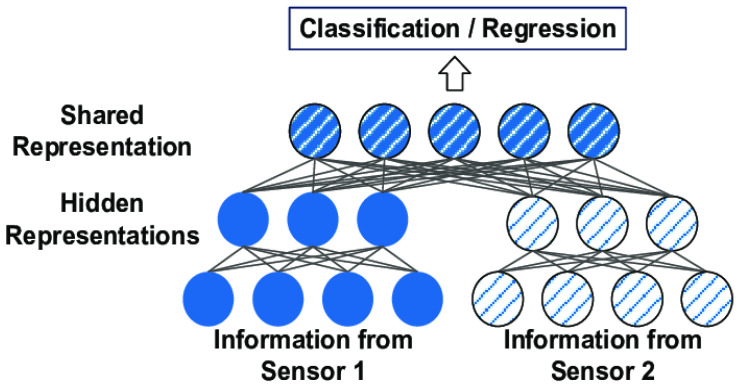
A baseline of multimodal fusion learning.

**Figure 2 sensors-21-04323-f002:**
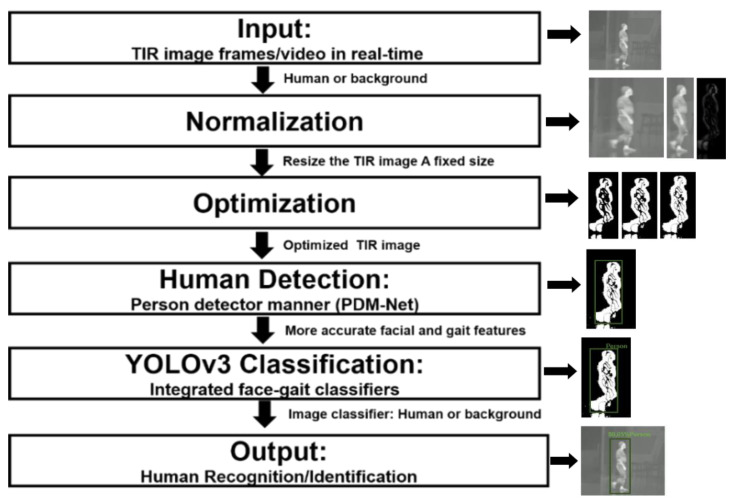
Flowchart of the proposed methodology procedure.

**Figure 3 sensors-21-04323-f003:**
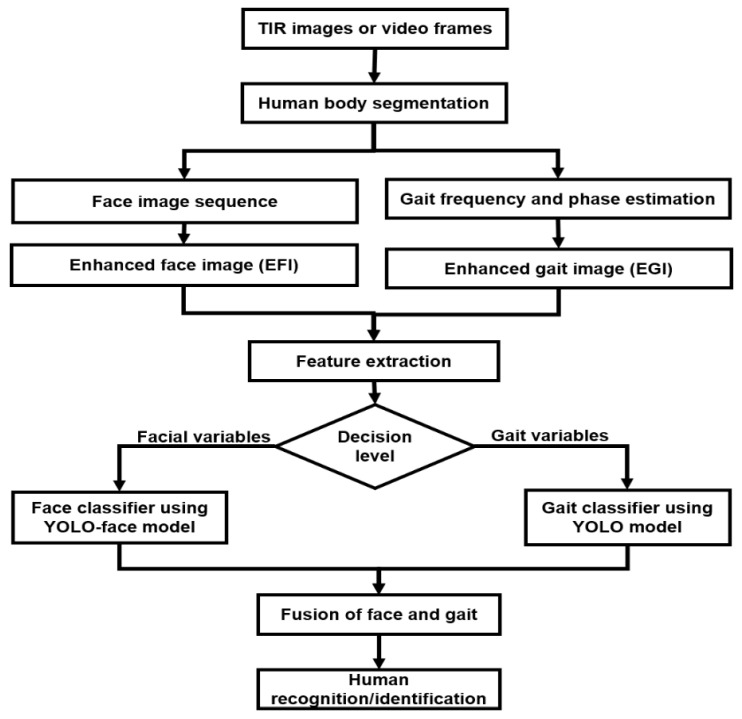
The overall tasks of multimodal (integrated face and gait classifiers) learning.

**Figure 4 sensors-21-04323-f004:**
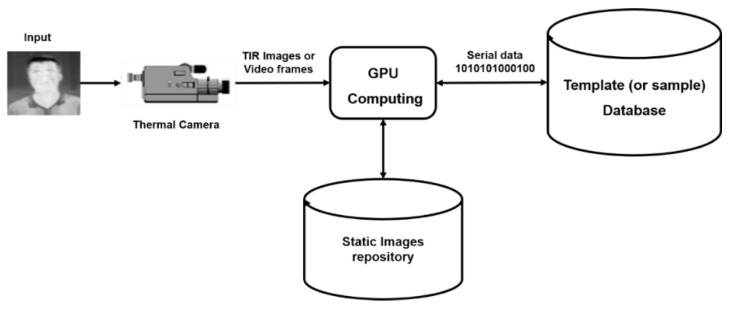
Face recognition processes.

**Figure 5 sensors-21-04323-f005:**
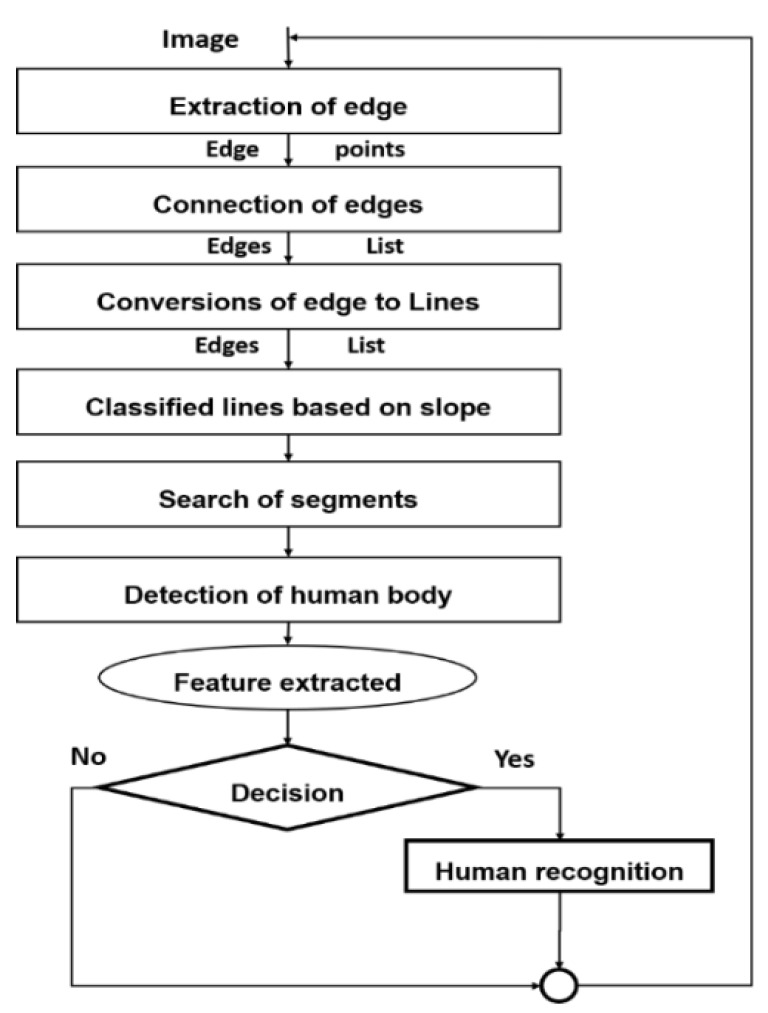
Gait recognition processes.

**Figure 6 sensors-21-04323-f006:**
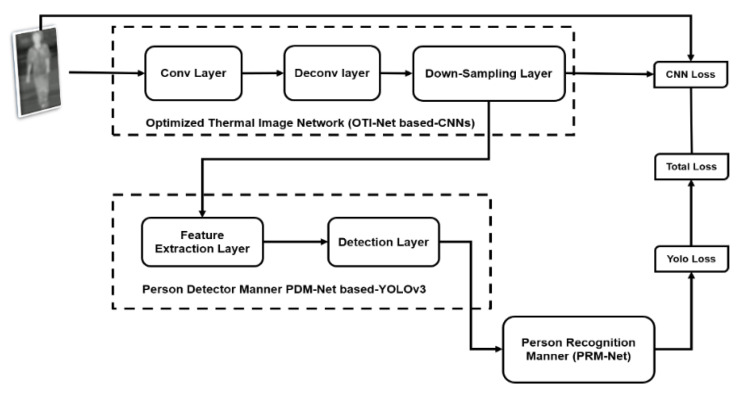
The block diagram of the proposed network.

**Figure 7 sensors-21-04323-f007:**
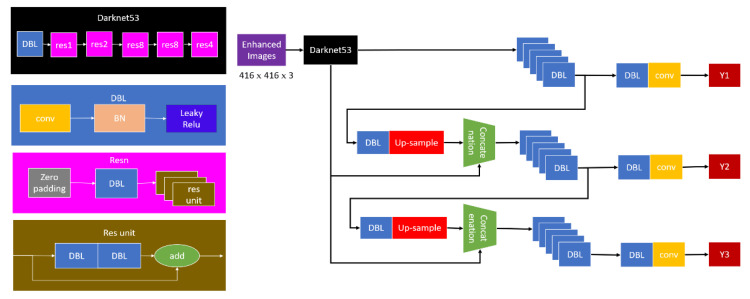
The PDM-Net architecture.

**Figure 8 sensors-21-04323-f008:**
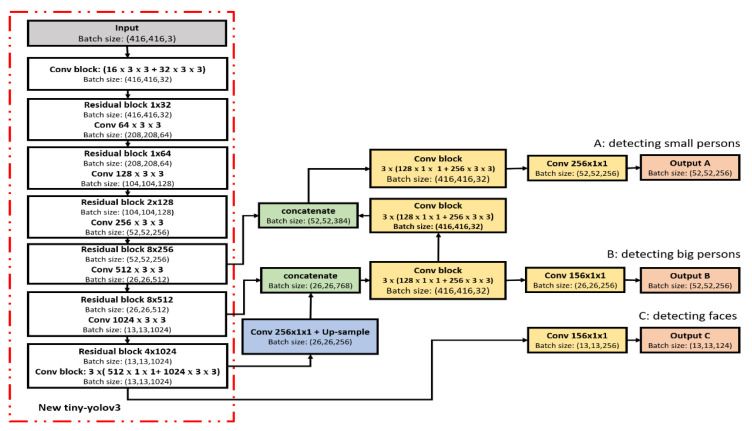
The new architecture of the tiny-yolov3.

**Figure 9 sensors-21-04323-f009:**
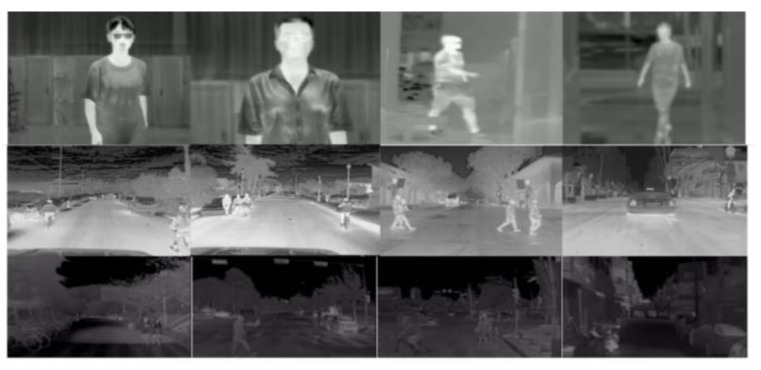
Example images from different night databases, Row 1: DHU Night dataset, Row 2: FLIR ADAS dataset, Row 3: KAIST dataset.

**Figure 10 sensors-21-04323-f010:**
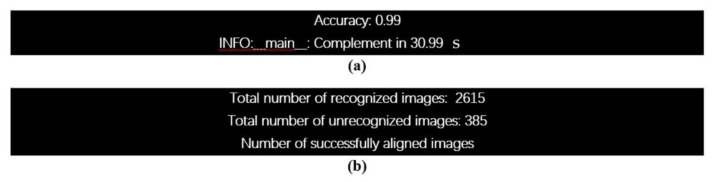
Evaluation test of the face classifiers on the DHU Night dataset for face recognition with gender classification: (**a**) accuracy rate, and (**b**) the result of the total number of recognized, unrecognized, and aligned images.

**Figure 11 sensors-21-04323-f011:**
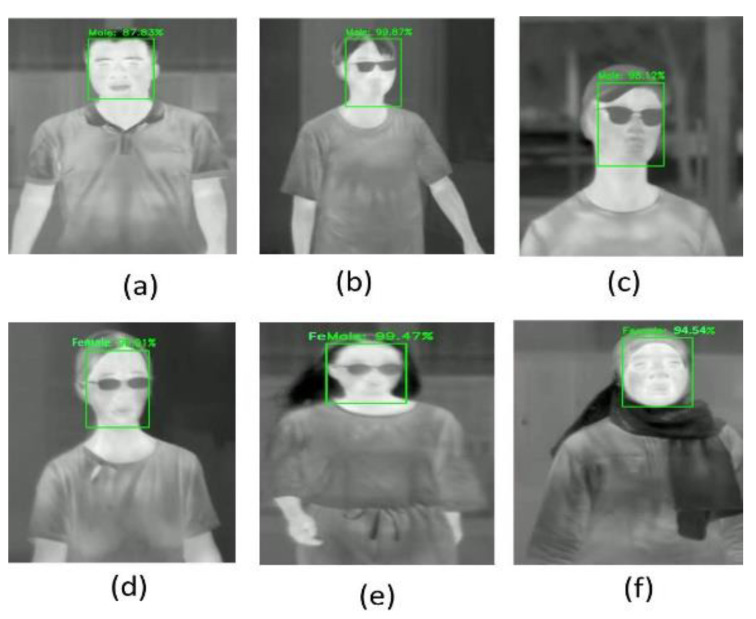
Evaluation test of the real-time face detector and gender classify in different scenarios on the DHU Night dataset. (**a**): Male, regular, (**b**): male with glasses, (**c**): male with glasses and a cap, (**d**): female with glasses and short hair, (**e**): female with glasses and long hair, (**f**): female with a scarf.

**Figure 12 sensors-21-04323-f012:**
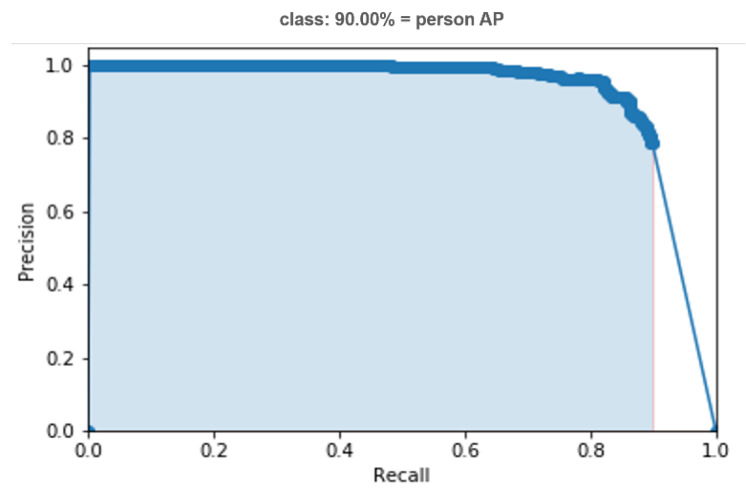
The AP score of PR curve for gait classifier trained on the DHU Night dataset.

**Figure 13 sensors-21-04323-f013:**
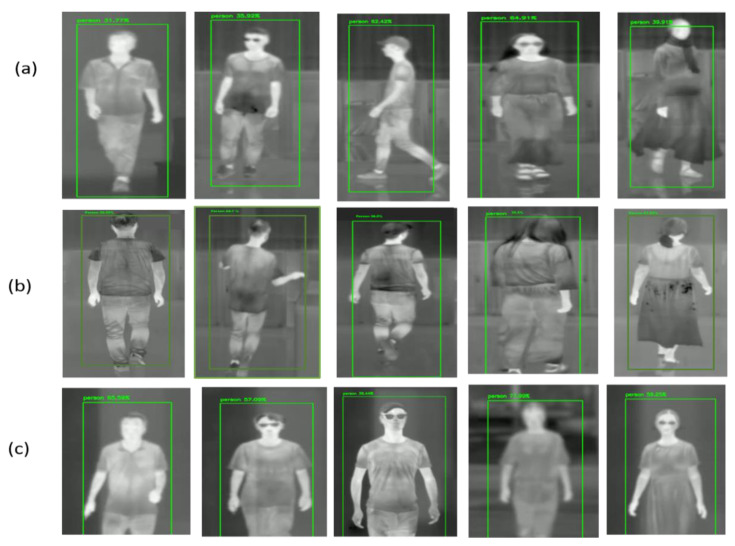
Evaluation tests of real-time night gait recognition on the DHU Night dataset for person classification: (**a**) the person recognition in regular walking conditions; (**b**) the person recognition in backside-walking conditions; (**c**) the person recognition using upper-body angles.

**Figure 14 sensors-21-04323-f014:**
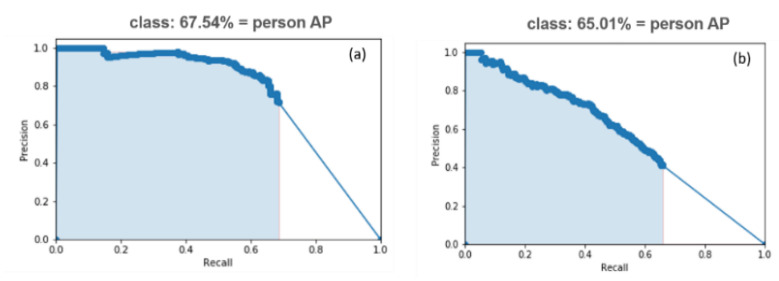
The AP score of PR curve for gait classifier trained on (**a**) FLIR ADAS dataset, and (**b**) KAIST dataset.

**Figure 15 sensors-21-04323-f015:**
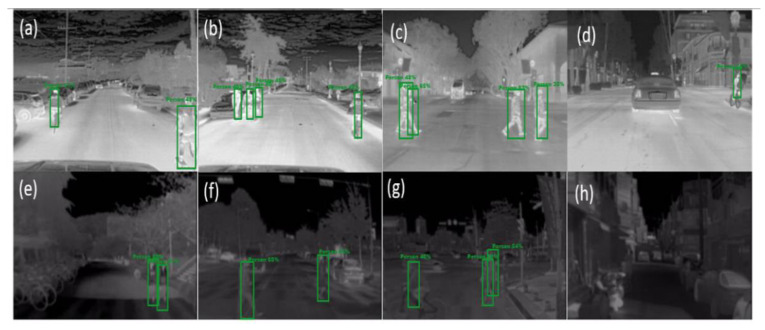
Qualitative results of human recognition (small object) on thermal images under different walking conditions, using our YOLOv3-Human model. Row 1: FLIR ADAS dataset. Row 2: KAIST dataset.

**Table 1 sensors-21-04323-t001:** Gait classifier recognition rates using individual variables and combinations of all spatiotemporal variables.

Gallery Size	Pelvis Angles		Hip Angles		Knee Angles		Ankle Angles		Foot Progress Angles	
	L	R	L	R	L	R	L	R	L	R
5	87	99	75	100	87	95	91	87	79	91
10	50	87	50	89	60	70	58	33	41	67

**Table 2 sensors-21-04323-t002:** Gait classifier recognition rates using lower-body angle trajectories.

Gallery Size	Spine Angles		Hip Angles		Knee Angles		Ankle Angles		Foot Progress Angles	
	L	R	L	R	L	R	L	R	L	R
5	87	99	75	100	87	95	91	87	79	91
10	50	87	50	89	60	70	58	33	41	67

**Table 3 sensors-21-04323-t003:** Gait classifier recognition rates using upper-body angle trajectories.

Gallery Size	Spine Angles		Neck Angles		Thorax Angles		Shoulder Angles		Elbow Angles	
	L	R	L	R	L	R	L	R	L	R
5	60	58	37	25	75	62	83	79	60	52
10	27	26	7	5	40	32	41	36	30	20

**Table 4 sensors-21-04323-t004:** Gait classifier recognition rates using various combinations of angle trajectories.

Gallery Size	Lower Body Angels	Upper Body Angels	All Angle Trajectories
5	95.83%	91.67%	96.50%
10	87.82%	82.69%	89.40%

**Table 5 sensors-21-04323-t005:** Performance comparison of the proposed YOLOv3-Human model with other methods on the same night databases.

Method	Database	Purpose	Detection Time	Accuracy (AP Score)
TIRFaceNet model	DHU dataset	Facial recognition		89.7%
YOLO model	FLIR dataset	Person detector for small and multiple subjects	-	29.36%
MMTOD model	KAIST dataset	Person detector for small and multiple subjects	-	49.39
	FLIR dataset		-	54.69
YOLOv3-Human model	KAIST dataset	Person detector for small and multiple subjects	10 ms	65.01%
	FLIR dataset		8 ms	67.54%
	DHU Night dataset	Integrated face and gait recognition	7 ms	99% for face and gender recognition 90% for gait recognition

## Data Availability

The DHU Night dataset presented in this study is available on request from the corresponding authors.

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
