# Peer review of "Real-Time Human Recognition at Night via Integrated Face and Gait Recognition Technologies"

_sensors, 2021, doi:10.3390/s21134323_

Round 1

Reviewer 1 Report

This paper proposes a novel approach for human recognition task in TIR images at night. The new network is based on YOLOv3 model by fusing face and gait classifiers to identify individuals automatically. The experimental results show that the proposed method is superior to others in accuracy and detection time.

The problems with the current version of the article are as follows:
1)In Section 3.1, the author claims that the variable size is much smaller than the square size adopted by YOLOv3, but the number given in brackets is greater than.

2)The author uses decision level fusion to fuse the results of face and gait, but lacks detailed description and formula for the fusion strategy.

3) The table of performance comparison is confusing. There are three types of purpose for different model. The author needs to explain how different purposes affect the final result and make a fair comparison of the human recognition task.

Reviewer 2 Report

The authors addressed the problem of human recognition at night. The manuscript is present an approach that combines face and gait to enhance the performance of real-time human recognition in TIR images at night considering several walking conditions. This approach is based on YOLOv3 and PRM-Net classifiers. The proposed method was evaluated using three datasets that simulate real surveillance conditions (DHU Night, FLIR, and KAIST) and achieve high accuracy results in comparison to other state-of-art approaches. In general, the document is well organized, but some revisions should be made to improve the quality of the paper:

1. The manuscript needs to be proofread to correct typos, grammatical errors and be more precise. Some sentences that need to improve are:

Line 56: sometimes not available enough
Line 111-112: the experiment tests were measured 
Line 116: Recently, customers see the most significant
Line 163: the inclusion of all different types of fun and exciting media in various fields
Line 184-185: (more detailed in subsection 2.2)
Line 218: (face or gait.) 
Line 481: set to10
Line 482: 20*30 epochs 
Line 499: Optimizer: we use the Darknet53 framework [54]
Line 531: (frame per second (FPS))
Line 542: are small or large sizes

2. Section 3.1 The Overall Procedure of the Proposed Methodology is hard to read. To improve the readability, it is suggested to the authors to split it into several paragraphs or use bullet points to identify the different steps.

3. Line 698-699: Rewrite the sentence “Moreover, the proposed model has a true positive (TP) for detecting small and multiple subjects on both FLIR and KAIST datasets.” It is not clear what the authors want to highlight?

4. In Figure 2, 3 and 5, it is suggested to the authors to include images to illustrate the steps of the workflows.

5. Why is used the 60-70% percentage of the dataset to train the Darknet model instead of a fixed percentage?

6. Why is choosing the tine-yolov3 architecture instead of the regular Yolov3 architecture that has a better performance?

7. Which trajectories are finally chosen for the gait recognition?

8. Why the performance of the gait classifier has a large decrease on the FLIR ADAS dataset (AP of 67.54%) and the KAIST dataset (AP of 65.01%) in comparison to the performance on the DHU Night dataset (AP of 90%)?

Reviewer 3 Report

The topic of real time human recognition at night sounds very interesting and challenging. Authors presented their proposed method quiet well, but some points should be improved:

  • Lines 59-60: some further reference could be useful.
  • It seems to be not so clear the usefulness of gender recognition in the proposed application. Could the author clarify the reason why they introduced gender recognition in addition to face recognition?
  • Concerning the gender recognition, it seems to be not clear where it occurs (wrt the overall tasks of multimodal learning)
  • Lines 318-323: these lines are not so clear. Is the system intended for face recognition, face verification or both?
  • Line 345: “[…] a torso has a central role in human recognition”. Authors should add some references to support this concept; torso measurement does not sound as very discriminating for subject recognition. Moreover, authors should better explain this paragraph (345-357) since it seems to be a little confused about body measurement and gait.
  • English language editing seems to be required.

Round 2

Reviewer 1 Report

The authors have carefully considered the suggestions and made a satisfied revision. I think it can be accepted.